# A Multiplicative Model for Learning Distributed Text-Based Attribute Representations

**Ryan Kiros, Richard S. Zemel, Ruslan Salakhutdinov**
University of Toronto
Canadian Institute for Advanced Research
{rkiros, zemel, rsalakhu}@cs.toronto.edu

## Abstract

In this paper we propose a general framework for learning distributed representations of attributes: characteristics of text whose representations can be jointly learned with word embeddings. Attributes can correspond to a wide variety of concepts, such as document indicators (to learn sentence vectors), language indicators (to learn distributed language representations), meta-data and side information (such as the age, gender and industry of a blogger) or representations of authors. We describe a third-order model where word context and attribute vectors interact multiplicatively to predict the next word in a sequence. This leads to the notion of conditional word similarity: how meanings of words change when conditioned on different attributes. We perform several experimental tasks including sentiment classification, cross-lingual document classification, and blog authorship attribution. We also qualitatively evaluate conditional word neighbours and attribute-conditioned text generation.

## 1 Introduction

Distributed word representations have enjoyed success in several NLP tasks [1, 2]. More recently, the use of distributed representations have been extended to model concepts beyond the word level, such as sentences, phrases and paragraphs [3, 4, 5, 6], entities and relationships [7, 8] and embeddings of semantic categories [9, 10].

In this paper we propose a general framework for learning distributed representations of attributes: characteristics of text whose representations can be jointly learned with word embeddings. The use of the word attribute in this context is general. Table 1 illustrates several of the experiments we perform along with the corresponding notion of attribute. For example, an attribute can represent an indicator of the current sentence or language being processed. This allows us to learn sentence and language vectors, similar to the proposed model of [6]. Attributes can also correspond to side information, or metadata associated with text. For instance, a collection of blogs may come with information about the age, gender or industry of the author. This allows us to learn vectors that can capture similarities across metadata based on the associated body of text. The goal of this work is to show that our notion of attribute vectors can achieve strong performance on a wide variety of NLP related tasks. In particular, we demonstrate strong quantitative performance on three highly diverse tasks: sentiment classification, cross-lingual document classification, and blog authorship attribution.

To capture these kinds of interactions between attributes and text, we propose to use a third-order model where attribute vectors act as gating units to a word embedding tensor. That is, words are represented as a tensor consisting of several prototype vectors. Given an attribute vector, a word embedding matrix can be computed as a linear combination of word prototypes weighted by the attribute representation. During training, attribute vectors reside in a separate lookup table which can be jointly learned along with word features and the model parameters. This type of three-way

Table 1: Summary of tasks and attribute types used in our experiments. The first three are quantitative while the second three are qualitative.

| Task | Dataset | Attribute type |
|---|---|---|
| Sentiment Classification | Sentiment Treebank | Sentence Vector |
| Cross-Lingual Classification | RCV1/RCV2 | Language Vector |
| Authorship Attribution | Blog Corpus | Author Metadata |
| Conditional Text Generation | Gutenberg Corpus | Book Vector |
| Structured Text Generation | Gutenberg Corpus | Part of Speech Tags |
| Conditional Word Similarity | Blogs & Europarl | Author Metadata / Language |

interaction can be embedded into a neural language model, where the three-way interaction consists of the previous context, the attribute and the score (or distribution) of the next word after the context.

Using a word embedding tensor gives rise to the notion of conditional word similarity. More specifically, the neighbours of word embeddings can change depending on which attribute is being conditioned on. For example, the word 'joy' when conditioned on an author with the industry attribute 'religion' appears near 'rapture' and 'god' but near 'delight' and 'comfort' when conditioned on an author with the industry attribute 'science'. Another way of thinking of our model would be the language analogue of [11]. They used a factored conditional restricted Boltzmann machine for modelling motion style defined by real or continuous valued style variables. When our factorization is embedded into a neural language model, it allows us to generate text conditioned on different attributes in the same manner as [11] could generate motions from different styles. As we show in our experiments, if attributes are represented by different books, samples generated from the model learn to capture associated writing styles from the author. Furthermore, we demonstrate a strong performance gain for authorship attribution when conditional word representations are used.

Multiplicative interactions have also been previously incorporated into neural language models. [12] introduced a multiplicative model where images are used for gating word representations. Our framework can be seen as a generalization of [12] and in the context of their work an attribute would correspond to a fixed representation of an image. [13] introduced a multiplicative recurrent neural network for generating text at the character level. In their model, the character at the current timestep is used to gate the network's recurrent matrix. This led to a substantial improvement in the ability to generate text at the character level as opposed to a non-multiplicative recurrent network.

## 2 Methods

In this section we describe the proposed models. We first review the log-bilinear neural language model of [14] as it forms the basis for much of our work. Next, we describe a word embedding tensor and show how it can be factored and introduced into a multiplicative neural language model. This is concluded by detailing how our attribute vectors are learned.

### 2.1 Log-bilinear neural language models

The log-bilinear language model (LBL) [14] is a deterministic model that may be viewed as a feed-forward neural network with a single linear hidden layer. Each word $w$ in the vocabulary is represented as a $K$-dimensional real-valued vector $\mathbf{r}_w \in \mathbb{R}^K$. Let $\mathbf{R}$ denote the $V \times K$ matrix of word representation vectors where $V$ is the vocabulary size. Let $(w_1, \ldots w_{n-1})$ be a tuple of $n-1$ words where $n-1$ is the context size. The LBL model makes a linear prediction of the next word representation as

$$\hat{\mathbf{r}} = \sum_{i=1}^{n-1} \mathbf{C}^{(i)} \mathbf{r}_{w_i}, \tag{1}$$

where $\mathbf{C}^{(i)}, i = 1, \ldots, n-1$ are $K \times K$ context parameter matrices. Thus, $\hat{\mathbf{r}}$ is the predicted representation of $\mathbf{r}_{w_n}$. The conditional probability $P(w_n = i | w_{1:n-1})$ of $w_n$ given $w_1, \ldots, w_{n-1}$ is

$$P(w_n = i | w_{1:n-1}) = \frac{\exp(\hat{\mathbf{r}}^T \mathbf{r}_i + b_i)}{\sum_{j=1}^{V} \exp(\hat{\mathbf{r}}^T \mathbf{r}_j + b_j)}, \tag{2}$$

where $\mathbf{b} \in \mathbb{R}^V$ is a bias vector. Learning can be done using backpropagation.

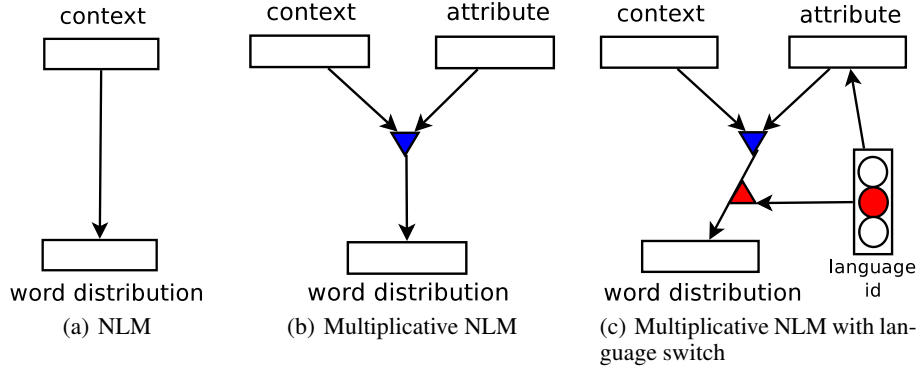

(a) NLM        (b) Multiplicative NLM        (c) Multiplicative NLM with language switch

Figure 1: Three different formulations for predicting the next word in a neural language model. **Left:** A standard neural language model (NLM). **Middle:** The context and attribute vectors interact via a multiplicative interaction. **Right:** When words are unshared across attributes, a one-hot attribute vector gates the factors-to-vocabulary matrix.

## 2.2 A word embedding tensor

Traditionally, word representation matrices are represented as a matrix $\mathbf{R} \in \mathbb{R}^{V \times K}$, such as in the case of the log-bilinear model. Throughout this work, we instead represent words as a tensor $\mathcal{T} \in \mathbb{R}^{V \times K \times D}$ where $D$ corresponds to the number of tensor slices. Given an attribute vector $\mathbf{x} \in \mathbb{R}^{D}$, we can compute attribute-gated word representations as $\mathcal{T}^{x} = \sum_{i=1}^{D} x_i \mathcal{T}^{(i)}$ i.e. word representations with respect to $\mathbf{x}$ are computed as a linear combination of slices weighted by each component $x_i$ of $\mathbf{x}$.

It is often unnecessary to use a fully unfactored tensor. Following [15, 16], we re-represent $\mathcal{T}$ in terms of three matrices $\mathbf{W}^{fk} \in \mathbb{R}^{F \times K}$, $\mathbf{W}^{fd} \in \mathbb{R}^{F \times D}$ and $\mathbf{W}^{fv} \in \mathbb{R}^{F \times V}$, such that

$$\mathcal{T}^{x} = (\mathbf{W}^{fv})^{\top} \cdot \text{diag}(\mathbf{W}^{fd}\mathbf{x}) \cdot \mathbf{W}^{fk}, \tag{3}$$

where $\text{diag}(\cdot)$ denotes the matrix with its argument on the diagonal. These matrices are parametrized by a pre-chosen number of factors $F$.

## 2.3 Multiplicative neural language models

We now show how to embed our word representation tensor $\mathcal{T}$ into the log-bilinear neural language model. Let $\mathbf{E} = (\mathbf{W}^{fk})^{\top}\mathbf{W}^{fv}$ denote a 'folded' $K \times V$ matrix of word embeddings. Given the context $w_1, \ldots, w_{n-1}$, the predicted next word representation $\hat{\mathbf{r}}$ is given by

$$\hat{\mathbf{r}} = \sum_{i=1}^{n-1} \mathbf{C}^{(i)} \mathbf{E}(:, w_i), \tag{4}$$

where $\mathbf{E}(:, w_i)$ denotes the column of $\mathbf{E}$ for the word representation of $w_i$ and $\mathbf{C}^{(i)}, i = 1, \ldots, n-1$ are $K \times K$ context matrices. Given a predicted next word representation $\hat{\mathbf{r}}$, the factor outputs are

$$\mathbf{f} = (\mathbf{W}^{fk}\hat{\mathbf{r}}) \bullet (\mathbf{W}^{fd}\mathbf{x}), \tag{5}$$

where $\bullet$ is a component-wise product. The conditional probability $P(w_n = i | w_{1:n-1}, \mathbf{x})$ of $w_n$ given $w_1, \ldots, w_{n-1}$ and $\mathbf{x}$ can be written as

$$P(w_n = i | w_{1:n-1}, \mathbf{x}) = \frac{\exp\big((\mathbf{W}^{fv}(:, i))^{\top}\mathbf{f} + b_i\big)}{\sum_{j=1}^{V} \exp\big((\mathbf{W}^{fv}(:, j))^{\top}\mathbf{f} + b_j\big)}.$$

Here, $\mathbf{W}^{fv}(:, i)$ denotes the column of $\mathbf{W}^{fv}$ corresponding to word $i$. In contrast to the log-bilinear model, the matrix of word representations $\mathbf{R}$ from before is replaced with the factored tensor $\mathcal{T}$, as shown in Fig. 1.

## 2.4 Unshared vocabularies across attributes

Our formulation for $\mathcal{T}$ assumes that word representations are shared across all attributes. In some cases, words may only be specific to certain attributes and not others. An example of this is cross-lingual modelling, where it is necessary to have language specific vocabularies. As a running example, consider the case where each attribute corresponds to a language representation vector. Let

Table 2: Samples generated from the model when conditioning on various attributes. For the last example, we condition on the average of the two vectors (symbol <#> corresponds to a number).

| Attribute | Sample |
|---|---|
| Bible | *<#> : <#> for thus i enquired unto thee , saying , the lord had not come unto him . <#> : <#> when i see them shall see me greater am that under the name of the king on israel .* |
| Caesar | *to tell vs pindarus : shortly pray , now hence , a word . comes hither , and let vs exclaim once by him fear till loved against caesar . till you are now which have kept what proper deed there is an ant ? for caesar not wise cassi* |
| $\frac{1}{2}$ (Bible + Caesar) | *let our spring tiger as with less ; for tucking great fellowes at ghosts of broth . industrious time with golden glory employments . <#> : <#> but are far in men soft from bones , assur too , set and blood of smelling , and there they cost , i learned : love no guile his word downe the mystery of possession* |

$\mathbf{x}$ denote the attribute vector for language $\ell$ and $\mathbf{x}'$ for language $\ell'$ (e.g. English and French). We can then compute language-specific word representations $\mathcal{T}^\ell$ by breaking up our decomposition into language dependent and independent components (see Fig. 1c):

$$\mathcal{T}^\ell = (\mathbf{W}_\ell^{fv})^\top \cdot \text{diag}(\mathbf{W}^{fd}\mathbf{x}) \cdot \mathbf{W}^{fk}, \tag{6}$$

where $(\mathbf{W}_\ell^{fv})^\top$ is a $V_\ell \times F$ language specific matrix. The matrices $\mathbf{W}^{fd}$ and $\mathbf{W}^{fk}$ do not depend on the language or the vocabulary, whereas $(\mathbf{W}_\ell^{fv})^\top$ is language specific. Moreover, since each language may have a different sized vocabulary, we use $V_\ell$ to denote the vocabulary size of language $\ell$. Observe that this model has an interesting property in that it allows us to share statistical strength across word representations of different languages. In particular, we show in our experiments how we can improve cross-lingual classification performance between English and German when a large amount of parallel data exists between English and French and only a small amount of parallel data exists between English and German.

## 2.5 Learning attribute representations

We now discuss how to learn representation vectors $\mathbf{x}$. Recall that when training neural language models, the word representations of $w_1, \ldots, w_{n-1}$ are updated by backpropagating through the word embedding matrix. We can think of this as being a linear layer, where the input to this layer is a one-hot vector with the $i$-th position active for word $w_i$. Then multiplying this vector by the embedding matrix results in the word vector for $w_i$. Thus the columns of the word representations matrix consisting of words from $w_1, \ldots, w_{n-1}$ will have non-zero gradients with respect to the loss. This allows us to consistently modify the word representations throughout training.

We construct attribute representations in a similar way. Suppose that $\mathbf{L}$ is an attribute lookup table, where $\mathbf{x} = f(\mathbf{L}(:, x))$ and $f$ is an optional non-linearity. We often use a rectifier non-linearity in order to keep $\mathbf{x}$ sparse and positive, which we found made training much more stable. Initially, the entries of $\mathbf{L}$ are generated randomly. During training, we treat $\mathbf{L}$ in the same way as the word embedding matrix. This way of learning language representations allows us to measure how 'similar' attributes are as opposed to using a one-hot encoding of attributes for which no such similarity could be computed.

In some cases, attributes that are available during training may not also be available at test time. An example of this is when attributes are used as sentence indicators for learning representations of sentences. To accommodate for this, we use an inference step similar to that proposed by [6]. That is, at test time all the network parameters are fixed and stochastic gradient descent is used for inferring the representation of an unseen attribute vector.

## 3 Experiments

In this section we describe our experimental evaluation and results. Throughout this section we refer to our model as Attribute Tensor Decomposition (ATD). All models are trained using stochastic gradient descent with an exponential learning rate decay and linear (per epoch) increase in momentum.

We first demonstrate initial qualitative results to get a sense of the tasks our model can perform. For these, we use the small project Gutenberg corpus which consists of 18 books, some of which have the same author. We first trained a multiplicative neural language model with a context size of 5,

Table 3: A modified version of the game Mad Libs. Given an initialization, the model is to generate the next 5 words according to the part-of-speech sequence (note that these are not hard constraints).

| [DT, NN, IN, DT, JJ] **the meaning of life is...** | [TO, VB, VBD, JJS, NNS] **my greatest accomplishment is...** | [PRP, NN, ',' , JJ, NN] **i could not live without...** |
|---|---|---|
| the cure of the bad | to keep sold most wishes | his regard , willing tenderness |
| the truth of the good | to make manned most magnificent | her french , serious friend |
| a penny for the fourth | to keep wounded best nations | her father , good voice |
| the globe of those modern | to be allowed best arguments | her heart , likely beauty |
| all man upon the same | to be mentioned most people | her sister , such character |

Table 4: Classification accuracies on various tasks. Left: Sentiment classification on the tree-bank dataset. Competing methods include the Neural Bag of words (NBoW) [5], Recursive Network (RNN) [17], Matrix-Vector Recursive Network (MV-RNN) [18], Recursive Tensor Network (RTNN) [3], Dynamic Convolutional Network (DCNN) [5] and Paragraph Vector (PV) [6]. Right: Cross-lingual classification on RCV2. Methods include statistical machine translation (SMT), I-Matrix [19], Bag-of-words autoencoders (BAE-*) [20] and BiCVM, BiCVM+ [21]. The use of '+' on cross-lingual tasks indicate the use of a third language (French) for learning embeddings.

| Method | Fine-grained | Positive / Negative | | Method | EN $\rightarrow$ DE | DE $\rightarrow$ EN |
|---|---|---|---|---|---|---|
| SVM | 40.7% | 79.4% | | SMT | 68.1% | 67.4% |
| BiNB | 41.9% | 83.1% | | I-Matrix | 77.6% | 71.1% |
| NBoW | 42.4% | 80.5% | | BAE-cr | 78.2% | 63.6% |
| RNN | 43.2% | 82.4% | | BAE-tree | 80.2% | 68.2% |
| MVRNN | 44.4% | 82.9% | | BiCVM | 83.7% | 71.4% |
| RTNN | 45.7% | 85.4% | | BiCVM+ | 86.2% | **76.9%** |
| DCNN | 48.5% | 86.8% | | BAE-corr | **91.8%** | 72.8% |
| PV | **48.7%** | **87.8%** | | ATD | 80.8% | 71.8% |
| ATD | 45.9% | 83.3% | | ATD+ | 83.4% | 72.9% |

where each attribute is represented as a book. This results in 18 learned attribute vectors, one for each book. After training, we can condition on a book vector and generate samples from the model. Table 2 illustrates some the generated samples. Our model learns to capture the 'style' associated with different books. Furthermore, by conditioning on the average of book representations, the model can generate reasonable samples that represent a hybrid of both attributes, even though such attribute combinations were not observed during training.

Next, we computed POS sequences from sentences that occur in the training corpus. We trained a multiplicative neural language model with a context size of 5 to predict the next word from its context, given knowledge of the POS tag for the next word. That is, we model $P(w_n = i|w_{1:n-1}, \mathbf{x})$ where $\mathbf{x}$ denotes the POS tag for word $w_n$. After training, we gave the model an initial input and a POS sequence and proceeded to generate samples. Table 3 shows some results for this task. Interestingly, the model can generate rather funny and poetic completions to the initial context.

## 3.1 Sentiment classification

Our first quantitative experiments are performed on the sentiment treebank of [3]. A common challenge for sentiment classification tasks is that the global sentiment of a sentence need not correspond to local sentiments exhibited in sub-phrases of the sentence. To address this issue, [3] collected annotations from the movie reviews corpus of [22] of all subphrases extracted from a sentence parser. By incorporating local sentiment into their recursive architectures, [3] was able to obtain significant performance gains with recursive networks over bag of words baselines.

We follow the same experimental procedure proposed by [3] for which evaluation is reported on two tasks: fine-grained classification of categories {very negative, negative, neutral, positive, very positive } and binary classification {positive, negative }. We extracted all subphrases of sentences that occur in the training set and used these to train a multiplicative neural language model. Here, each attribute is represented as a sentence vector, as in [6]. In order to compute subphrases for unseen sentences, we apply an inference procedure similar to [6], where the weights of the network are frozen and gradient descent is used to infer representations for each unseen vector. We trained a logistic regression classifier using all training subphrases in the training set. At test time, we infer a representation for a new sentence which is used for making a review prediction. We used a context

size of 8, 100 dimensional word vectors initialized from [2] and 100 dimensional sentence vectors initialized by averaging vectors of words from the corresponding sentence.

Table 4, left panel, illustrates our results on this task in comparison to all other proposed approaches. Our results are on par with the highest performing recursive network on the fine-grained task and outperforms all bag-of-words baselines and recursive networks with the exception of the RTNN on the binary task. Our method is outperformed by the two recently proposed approaches of [5] (a convolutional network trained on sentences) and Paragraph Vector [6].

## 3.2   Cross-lingual document classification

We follow the experimental procedure of [19], for which several existing baselines are available to compare our results. The experiment proceeds as follows. We first use the Europarl corpus [23] for inducing word representations across languages. Let $S$ be a sentence with words $w$ in language $\ell$ and let $\mathbf{x}$ be the corresponding language vector. Let

$$v_\ell(S) = \sum_{w \in S} \boldsymbol{\mathcal{T}}^\ell(:, w) = \sum_{w \in S} (\mathbf{W}_\ell^{fv}(:, w))^\top \cdot \operatorname{diag}(\mathbf{W}^{fd}\mathbf{x}) \cdot \mathbf{W}^{fk} \tag{7}$$

denote the sentence representation of $S$, defined as the sum of language conditioned word representations for each $w \in S$. Equivalently we define a sentence representation for the translation $S'$ of $S$ denoted as $v_{\ell'}(S')$. We then optimize the following ranking objective:

$$\underset{\theta}{\text{minimize}} \quad \sum_S \sum_k \max \left\{ 0, \alpha + \left\| v_\ell(S) - v_{\ell'}(S') \right\|_2^2 - \left\| v_\ell(S) - v_{\ell'}(C_k) \right\|_2^2 \right\} + \lambda \|\theta\|_2^2$$

subject to the constraints that each sentence vector has unit norm. Each $C_k$ is a constrastive (non-translation) sentence of $S$ and $\theta$ denotes all model parameters. This type of cross-language ranking loss was first used by [21] but without the norm constraint which we found significantly improved the stability of training. The Europarl corpus contains roughly 2 million parallel sentence pairs between English and German as well as English and French, for which we induce 40 dimensional word representations. Evaluation is then performed on English and German sections of the Reuters RCV1/RCV2 corpora. Note that these documents are not parallel. The Reuters dataset contains multiple labels for each document. Following [19], we only consider documents which have been assigned to one of the top 4 categories in the label hierarchy. These are CCAT (Corporate/Industrial), ECAT (Economics), GCAT (Government/Social) and MCAT (Markets). There are a total of 34,000 English documents and 42,753 German documents with vocabulary sizes of 43614 English words and 50,110 German words. We consider both training on English and evaluating on German and vice versa. To represent a document, we sum over the word representations of words in that document followed by a unit-ball projection. Following [19] we use an averaged perceptron classifier. Classification accuracy is then evaluated on a held-out test set in the other language. We used a monolingual validation set for tuning the margin $\alpha$, which was set to $\alpha = 1$. Five contrastive terms were used per example which were randomly assigned per epoch.

Table 4, right panel, shows our results compared to all proposed methods thus far. We are competitive with the current state-of-the-art approaches, being outperformed only by BiCVM+ [21] and BAE-corr [20] on EN $\rightarrow$ DE. The BAE-corr method combines both a reconstruction term and a correlation regularizer to match sentences, while our method does not consider reconstruction. We also performed experimentation on a low resource task, where we assume the same conditions as above with the exception that we only use 10,000 parallel sentence pairs between English and German while still incorporating all English and French parallel sentences. For this task, we compare against a separation baseline, which is the same as our model but with no parameter sharing across languages (and thus resembles [21]). Here we achieve 74.7% and 69.7% accuracies (EN→DE and DE→EN) while the separation baseline obtains 63.8% and 67.1%. This indicates that parameter sharing across languages can be useful when only a small amount of parallel data is available. Figure 2 further shows $t$-SNE embeddings of English-German word pairs.[1]

Another interesting consideration is whether or not the learned language vectors can capture any interesting properties of various languages. To look into this, we trained a multiplicative neural language model simultaneously on 5 languages: English, French, German, Czech and Slovak. To our knowledge, this is the most languages word representations have been jointly learned on. We

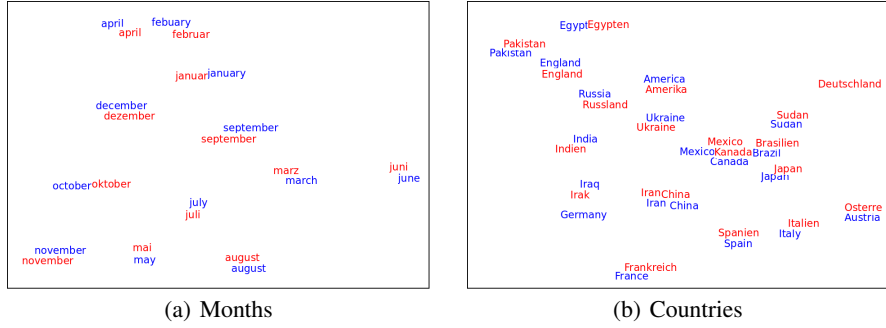

(a) Months                                    (b) Countries

Figure 2: t-SNE embeddings of English-German word pairs learned from Europarl.

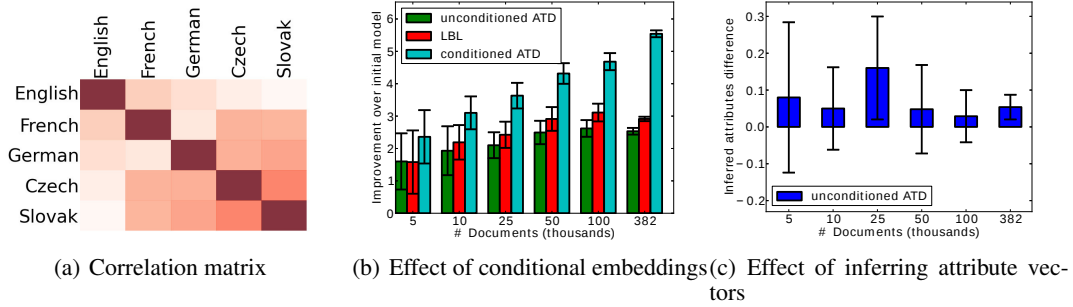

(a) Correlation matrix        (b) Effect of conditional embeddings  (c) Effect of inferring attribute vectors

Figure 3: Results on the Blog classification corpus. For the middle and right plots, each pair of same coloured bars corresponds to the non-inclusion or inclusion of inferred attribute vectors, respectively.

computed a correlation matrix from the language vectors, illustrated in Fig. 3a. Interestingly, we observe high correlation between Czech and Slovak representations, indicating that the model may have learned some notion of lexical similarity. That being said, additional experimentation for future work is necessary to better understand the similarities exhibited through language vectors.

### 3.3 Blog authorship attribution

For our final task, we use the Blog corpus of [24] which contains 681,288 blog posts from 19,320 authors. For our experiments, we break the corpus into two separate datasets: one containing the 1000 most prolific authors (most blog posts) and the other containing all the rest. Each author comes with an attribute tag corresponding to a tuple (age, gender, industry) indicating the age range of the author (10s, 20s or 30s), whether the author is male or female, and what industry the author works in. Note that industry does not necessary correspond to the topic of blog posts. We use the dataset of non-prolific authors to train a multiplicative language model conditioned on an attribute tuple of which there are 234 unique tuples in total. We used 100 dimensional word vectors initialized from [2], 100 dimensional attribute vectors with random initialization and a context size of 5. A 1000-way classification task is then performed on the prolific author subset and evaluation is done using 10-fold cross-validation. Our initial experimentation with baselines found that tf-idf performs well on this dataset (45.9% accuracy). Thus, we consider how much we can improve on the tf-idf baseline by augmenting word and attribute features.

For the first experiment, we determine the effect conditional word embeddings have on classification performance, assuming attributes are available at test time. For this, we compute two embedding matrices from a trained ATD model, one without and with attribute knowledge:

$$\text{unconditioned ATD}: \quad (\mathbf{W}^{fv})^\top \mathbf{W}^{fk} \tag{8}$$

$$\text{conditioned ATD}: \quad (\mathbf{W}^{fv})^\top \cdot \text{diag}(\mathbf{W}^{fd}\mathbf{x}) \cdot \mathbf{W}^{fk}. \tag{9}$$

We represent a blog post as the sum of word vectors projected to unit norm and augment these with tf-idf features. As an additional baseline we include a log-bilinear language model [14]. [2] Figure 3b illustrates the results from which we observe that conditioned word embeddings are significantly more discriminative over word embeddings computed without knowledge of attribute vectors.

Table 5: Results from a conditional word similarity task using Blog attributes and language vectors.

| Query,A,B | Common | Unique to A | Unique to B | | English | French | German |
|-----------|--------|-------------|-------------|---|---------|--------|--------|
| school | work | choir | therapy | | *january* | janvier | januar |
| f/10/student | church | prom | tech | | june | decembre | dezember |
| m/20/tech | college | skool | job | | october | juin | juni |
| journal | diary | project | zine | | *market* | marche | markt |
| f/10/student | blog | book | app | | markets | marches | binnenmarktes |
| m/30/adv. | webpage | yearbook | referral | | internal | interne | marktes |
| create | build | provide | compile | | *war* | guerre | krieg |
| f/30/arts | develop | acquire | follow | | weapons | terrorisme | globale |
| f/30/internet | maintain | generate | analyse | | global | mondaile | krieges |
| joy | happiness | rapture | delight | | *said* | dit | sagte |
| m/30/religion | sadness | god | comfort | | stated | disait | gesagt |
| m/20/science | pain | heartbreak | soul | | told | declare | sagten |
| cool | nice | beautiful | sexy | | *two* | deux | zwei |
| m/10/student | funny | amazing | hott | | two-thirds | deuxieme | beiden |
| f/10/student | awesome | neat | lame | | both | seconde | zweier |

For the second experiment, we determine the effect of inferring attribute vectors at test time if they are not assumed to be available. To do this, we train a logistic regression classifier within each fold for predicting attributes. We compute an inferred vector by averaging each of the attribute vectors weighted by the log-probabilities of the classifier. In Fig. 3c we plot the difference in performance when an inferred vector is augmented vs. when it is not. These results show consistent, albeit small improvement gains when attribute vectors are inferred at test time.

To get a better sense of the attribute features learned from the model, the supplementary material contains a t-SNE embedding of the learned attribute vectors. Interestingly, the model learns features which largely isolate the vectors of all teenage bloggers independent of gender and topic.

### 3.4 Conditional word similarity

One of the key properties of our tensor formulation is the notion of conditional word similarity, namely how neighbours of word representations change depending on the attributes that are conditioned on. In order to explore the effects of this, we performed two qualitative comparisons: one using blog attribute vectors and the other with language vectors. These results are illustrated in Table 5. For the first comparison on the left, we chose two attributes from the blog corpus and a query word. We identify each of these attribute pairs as A and B. Next, we computed a ranked list of the nearest neighbours (by cosine similarity) of words conditioned on each attribute and identified the top 15 words in each. Out of these 15 words, we display the top 3 words which are common to both ranked lists, as well as 3 words that are unique to a specific attribute. Our results illustrate that the model can capture distinctive notions of word similarities depending on which attributes are being conditioned. On the right of Table 5, we chose a query word in English (italicized) and computed the nearest neighbours when conditioned on each language vector. This results in neighbours that are either direct translations of the query word or words that are semantically similar. The supplementary material includes additional examples with nearest neighbours of collocations.

## 4 Conclusion

There are several future directions from which this work can be extended. One application area of interest is in learning representations of authors from papers they choose to review as a way of improving automating reviewer-paper matching [25]. Since authors contribute to different research topics, it might be more useful to instead consider a mixture of attribute vectors that can allow for distinctive representations of the same author across research areas. Another interesting application is learning representations of graphs. Recently, [26] proposed an approach for learning embeddings of nodes in social networks. Introducing network indicator vectors could allow us to potentially learn representations of full graphs. Finally, it would be interesting to train a multiplicative neural language model simultaneously across dozens of languages.

### Acknowledgments

We would also like to thank the anonymous reviewers for their valuable comments and suggestions. This work was supported by NSERC, Google, Samsung, and ONR Grant N00014-14-1-0232.

## Footnotes

[1]We note that Germany and Deutschland are nearest neighbours in the original space.

[2]The log-bilinear model has no concept of attributes.

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
