[Supplementary Material]

Table 1: Nearest neighbours of collocations. For each English query on the left, we condition on the language vector for English, French and German and retrieve the nearest collocations. Collocations are represented by summing the conditional word vectors and scaling them to unit norm.

| declare war | declare war on<br>guerre eclair<br>den drohenden krieg | to declare war<br>guerre<br>drohenden krieg | declares war<br>la guerre iran-irak<br>kalte krieg |
|---|---|---|---|
| private sector | private sector which<br>secteur prive<br>privaten sektor | private sector there<br>secteur prive dans<br>dem privaten sektor | private sector in<br>du secteur prive<br>den privaten sektor |
| one million euros | around one million<br>quelque EUR millions<br>anderthalb millionen | one hundred million<br>millions de dollars<br>millionen von dollar | million euros<br>de EUR millions<br>eine halbe millionen |
| united states | united states australia<br>etats-unis investissent<br>vereinigten staaten | united mexican states<br>etats-unis bloquent<br>den vereinigten staaten | united states takes<br>etats-unis<br>vereinigten staaten bei |
| first session | first meeting<br>premiere session<br>ersten wahlgang | first appearance<br>premiere phase<br>ersten auftritt | first anniversary of<br>premiere observation<br>ersten vizeprasidenten |
| european union | european union 1-6<br>union europeenne<br>zur europaischen union | european union spends<br>'union europeenne aa-afns<br>europaischen union | european union undertook<br>'union europeenne<br>europaische union |
| police officer | police<br>academie de police<br>polizei | police academy<br>agents de police<br>polizei militar | institute of police<br>officiers de police<br>kriminellen handlungen in |
| on monday | meeting on monday<br>au mois d'octobre<br>am montag vor | relating to monday<br>soiree de lundi<br>am montag mit | monday<br>lundi<br>montag am |
| south africa | south africa of<br>'afrique du sud<br>sudlichen afrika | of south africa<br>afrique du sud<br>sudlichen afrikas | from south africa<br>l'afrique du sud<br>sudliche afrika |
| poor performance | poor<br>pauvres<br>armen | poor levels<br>pauvres recoivent<br>hoch verschuldeten armen | bitterly poor<br>pauvres lourdement endettes<br>den armen |
| low income | low incomes<br>a faibles revenus<br>niedrigen einkommen | on low incomes<br>a faible revenu<br>geringen einkommen | lower incomes<br>faible revenu<br>landwirtschaftlichen einkommen |
| military invasion | military invasion of<br>composante militaire<br>militarischen | high-ranking military<br>sous occupation militaire<br>anschaffung von militarischen | military tribunals<br>attache militaire de<br>militarischen intervention |
| why we oppose | why we abstained<br>pourquoi nous votons<br>warum lehnen wir | reasons why we<br>pourquoi nous approuvons<br>grunde warum wir | why we<br>pourquoi nous lancons<br>warum schicken wir |
| church and state | state and non-state<br>comportements racistes et<br>korperliche und psychische | protectionism and state<br>'etat providence et<br>und psychische | rogue state and<br>raciales et ethniques<br>kirche und staat |
| islamic republic | islamic republic of<br>republique islamique<br>islamischen republik | islamic federal republic<br>en republique islamique<br>islamische republik iran | the islamic republic<br>republique islamique d<br>islamische republik |

Figure 1: $t$-SNE embedding of blog attributes. Each attribute vector is represented as a tuple (age range, gender, industry). Best viewed electronically.