[Reviews · NeurIPS 2014]

Submitted by Assigned_Reviewer_1

This paper proposes to incorporate side information for improving vector-space embedding of words via an "attribute vector" that modulates the word-projection matrices. One could simply think of word-projection tensors (although, in practice the tensors are factorized) where the attribute vector provide the loadings for the tensor slices. This is studied in the context of log-bilinear language models, but the basic idea should be applicable to other word embedding work.

The theory part of the paper is very well-written. However, it is in the experimental section that things get somewhat muddier. I would have preferred to see fewer experiments with greater depth than a barrage of experiments with little insight. Firstly, I am not sure what the experiments at the beginning of section 3 (contents of Tables 2 & 3) are really showing. Many models can generate text, and I am not sure these examples are showing anything in particular about the proposed model. Even a simple n-gram LM can interpolate between Bible and Caesar. Also, please mention what values of the initial learning rate, decay factor, momentum, and momentum increase factor were used.

In section 3.1, what are the attributes? Is is just a sentence vector that is an average of all sub-phrases, or something else? Could you motivate why the choice is reasonable as an attribute?

In section 3.2, please change the notation to make it clear that S is from l and S' and C_k are from l'. Perhaps v should really be v_l. Also, what are the attributes (x) in this case? In figure 1, right: you talk about a 1-hot attribute vector, but that's really the language-id vector and not the attribute vector. Also, do you have any insight as to why Germany and Deutschland are far apart while the translations of all other countries appear close together? Is it an error, some noise not accounted for, or showing something interesting about what the model learns (perhaps some us vs. them distinction, whereby Germany in English and Deutschland in German aren't learnt to be the same concept)?

In section 3.3, if there are 234 unique attributes, why do you use a 100-dimensional attribute vector?
Summary: The paper introduces the idea of an attribute vector to modulate the vector-space embedding of words. The theory is presented well, however the experiments could be improved.

Submitted by Assigned_Reviewer_23

The authors extend log-bilinear language models by replacing the traditional two-way interactions (matrix) with three-way interactions (tensor, which is then factorized). That is, the authors replace the energy with a factored energy. This is what is routinely done when transforming RBMs into a factored RBMs.

The idea of using factored models has been widely explored in a number of tasks and applications. Nonetheless, the authors present several nice examples of the application of factored log-bilinear models to language tasks: context sensitive language models, sentiment classification, cross-lingual document classification, blog authorship attribution, and conditional word similarity.

What I found most interesting in these applications was the choice of attributes. I was disappointed with the performance of the models on quantitative tasks. Here the authors point out that with sophisticated hyper-parameter search the gap can narrow. This I believe is speculative. It is also conceivable that these are relatively small datasets and consequently models with more parameters will need better regularizers (or much more data).

The experiment on blog authorship makes a good case for a wider adoption of factored log-bilinear models in language tasks that use log-bilinear models. The experiment on conditional word similarity is very nice.

Summary: The paper applies factored log-bilinear models to a wide range of language applications and demonstrates the value of context variables to some extent. The paper is well written, fun and clear.

Submitted by Assigned_Reviewer_32

This paper introduces a word representation model that can be conditioned on attributes.

The idea is simple, well described and illustrated with several examples.
Unfortunately, none of the experiments are very impressive and the model is a very simple delta from previous papers.

Marginally above the acceptance threshold
Summary: This paper introduces a word representation model that can be conditioned on attributes.

The idea is simple, well described and illustrated with several examples.
Unfortunately, none of the experiments are very impressive and the model is a very simple delta from previous papers.

Marginally above the acceptance threshold
Author Feedback
Author rebuttal: Thank you for your valuable feedback.

To all reviewers: We would like to first address the main concern, which is experimentation.

Our goal in this paper was to demonstrate versatility, across a wide range of experimental tasks. We felt this was a reasonable path of action due to the fact that our concept of attributes is itself very general and not restricted to any particular task. Moreover:

- We demonstrated an advantage in cross-lingual classification when parallel data is scarce. This was only mentioned in-line (lines 313-320) but we will
emphasize this more. Furthermore, all methods compared against for cross-lingual classification were designed specifically for the task of learning bilingual representations and are not directly applicable to other experiments in the paper.

- The blog authorship classification experiments demonstrated an improvement in the discriminative ability of word representations through attribute vectors, a task that naturally requires a 3rd-order model of words and is thus not directly applicable to any of the methods compared against in other experiments.

- Methods used for sentiment classification (e.g. RNNs, DCNNs) could arguably be adapted to cross-lingual classification but would require a non-trivial amount of modification. Our proposed method on the other hand only requires modifying the attribute concept and the loss function.

- All methods that outperform us were only published in the last few months (ICML 2014, ACL 2014, ICLR 2014). It is very challenging to both achieve versatility and consistent state-of-the-art results, particularly against models designed specifically and optimized for one particular task.

We feel our experiments have demonstrated strong generality across several diverse tasks, which is a natural goal given our model framework.

Addtional comments to Reviewer_1:

- Tables 2 & 3: These were included to show that the style of text can be directly modified by the choice of attribute vector being conditioned on. It also gives a visual sanity check, as this should be the expected behavior of the model.

- Hyperparameters: Details of the hyperparameters for all models will be included in the supplementary material.

- Section 3.1: The attributes are the representations of the sentences themselves. One can think of attribute indicators as being a type of 'label' for each instance in the training set. In the case of learning sentence representations, each training example thus gets its own label. It is a more extreme setting of the model and our use of it was motivated by the strong results obtained by [6].

- Notation: Thanks for the suggestion. We will modify this.

- Section 3.2: The attributes here act as representations for languages. The 1-hot language ID is used as a way of both doing table lookup to obtain the language vector as well as gate the vocabulary matrix for the language currently being used. When presenting the correlation matrix, it is these attribute vectors that are used for computation.

- Germany-Deutschland: This is actually a fault of t-SNE. In the original embedding space, Germany and Deutschland are nearest neighbours. We will clarify this.

- 100-dim vector: This is a good point. We did this just for consistency with other experiments but arguably it is not necessary to use a 100-dim vector on this dataset. We will clarify this.